# Evaporation-driven transport-control of small molecules along nanoslits

Sangjin Seo [1,2], Dogyeong Ha[1,2] & Taesung Kim [1✉]

Understanding and controlling the transport mechanisms of small molecules at the micro/nanoscales is vital because they provide a working principle for a variety of practical micro/nanofluidic applications. However, most precedent mechanisms still have remaining obstacles such as complicated fabrication processes, limitations of materials, and undesired damage on samples. Herein, we present the evaporation-driven transport-control of small molecules in gas-permeable and low-aspect ratio nanoslits, wherein both the diffusive and advective mass transports of solutes are affected by solvent evaporation through the nanoslit walls. The effect of the evaporation flux on the mass transport of small molecules in various nanoslit-integrated micro/nanofluidic devices is characterized, and dynamic transport along the nanoslit is investigated by conducting numerical simulations using the advection-diffusion equation. We further demonstrate that evaporation-driven, nanoslit-based transport-control can be easily applied to a micro/nanofluidic channel network in an independent and addressable array, offering a unique working principle for micro/nanofluidic applications and components such as molecule-valves, -concentrators, -pumps, and -filters.

---

[1] Department of Mechanical Engineering, Ulsan National Institute of Science and Technology (UNIST), 50 UNIST-gil, Ulsan, Republic of Korea. [2]These authors contributed equally: Sangjin Seo, Dogyeong Ha. ✉email: tskim@unist.ac.kr

The transport control of small molecules in micro/nano-fluids demonstrates remarkable potential for a variety of practical micro/nanofluidic devices involved in drug delivery[1], biosensing[2–4], energy conversion[5–7], and water desalination[8]. To date, most micro/nanofluidic devices have focused on external energy source-dependent approaches that are easy to employ and switch. For example, voltage-driven gates have been developed to control the liquid/vapor interface in hydrophobic, solid-state nanopores[9–14]. Functional materials have been utilized to develop alternative gates that use intrinsic material properties that respond to physiochemical stimuli, such as pH[15], light[16], and temperature[17]. Recently, functional, liquid-based gates have been reported that utilize their immiscibility with water[18–20]. Such gates that control small molecule transport in micro/nanofluidic devices rely on functional nanoporous membrane structures.

The transport capability of each of the abovementioned approaches must be assessed. First, voltage-driven gates are simply designed and fabricated using common materials; however, samples with electrical properties may be damaged at high electric potential or current. Second, functional material adoption is often limited because of the intrinsic material properties including side effects with target samples, availability of the fabrication process, and integration into the micro/nanofluidic devices. Third, more powerful gates are necessary to enable the active and versatile control of small molecules, such as valving, concentrating, pumping, and filtering abilities on a chip. To date, a versatile and multi-functional gate on a chip has not yet been demonstrated despite the remarkable potential for micro/nano-fluidic mass transport controllers.

Herein, we describe the evaporation-driven transport control of small molecules along nanoslits. We experimentally investigate evaporation-driven mass transport using the polydimethylsiloxane (PDMS) system that not only exhibits a nanoporous property but also enables the generation of an evaporation-driven flow[21,22]. In addition, we conduct numerical simulations by solving the advection–diffusion equation that considers the advective transport caused by solvent evaporation through the nanoslit walls. Furthermore, we demonstrate that the transport of small molecules in a micro/nanofluidic device can be actively manipulated such that the evaporation flux through the nanoslit is globally and locally manipulated.

## Results

**Global evaporation-control.** Figure 1a illustrates a global, evaporation-controlled, micro/nanofluidic device (GECMN), wherein two microchannels for solution loading with (i.e., source channel) and without (i.e., drain channel) solutes are connected with a nanoslit and a test chamber with a 100-μm diameter. We employed cracking-assisted photolithography to fabricate a GECMN integrated with a gas-permeable, PDMS-based nanoslit, which permitted diffusive mass transport but suppressed pressure-driven flow via high hydraulic resistance[23–26]. PDMS is a gas-permeable material, which enabled the water molecules to continuously penetrate the nanoslit walls and evaporate into air[27]. Therefore, the evaporation flux of the solvent was affected by the relative humidity (RH) of the global environment. The test chamber was fabricated as a region of interest to quantify the mass transport rate along the nanoslit by measuring the fluorescence intensity (FI).

Figure 1b depicts the working principle of the transport control of small molecules in a micro/nanofluidic device. Dry conditions dehydrated the PDMS device, such that the solvent molecules were absorbed from the nanoslit to the PDMS walls, and then evaporated into the air. Owing to the high surface to volume ratio of the nanoslit, the solvent evaporation generated advective flow

from the ends of the nanoslit toward the center. Furthermore, the advective transport was strong enough to compete with the diffusive transport of the solutes, such that the nanoslit acted as a gate manipulated by the evaporation flux, which in turn depended on the global RH. As a result, the solutes concentrated over time and were subject to diffusion from the center to the drain channel. Therefore, during dehydration, the diffusive mass transport of the solutes was interrupted from the center to the drain channel, resulting in a closed state. Conversely, the evaporation flux through the nanoslit walls was easily lowered by varying the conditions from dry to humid, corresponding to global humidity control (Fig. 1c). A fully hydrated GECMN prohibited evaporation in the nanoslit, eliminating the advective flow and allowing the accumulated solutes to diffuse toward the source and drain channels. After the solute distribution in the nanoslit reached a steady state, the diffusive transport of the solutes occurred while the advective transport ceased.

Figure 1d depicts the microscopy and atomic force microscopy (AFM) images of the micro/nanofluidic device with a single nanoslit (~3-μm wide, 200-nm deep). The black dashed square represents the test chamber. Figure 1e illustrates microscopic images of the nanoslit during hydration and dehydration; the length of the nanoslit was defined as $2L = 800$ μm. After soaking the GECMN in water for 24 h, a buffer solution was loaded with small water-soluble molecules (i.e., fluorescein isothiocyanate (FITC)) in the source channel to observe the transport behavior inside the nanoslit by measuring the FI. During hydration, the nanoslit was clearly observed because it was filled with the solution. During dehydration, the nanoslit center was blurred. Light diffraction occurred, due to the supersaturation of the small molecules. Reversible switching occurred between dehydration and rehydration, indicating closed- and open-gating of the solute transport (Fig. 1f). During dehydration for 1 h (RH ~20%), the continuous advective flow concentrated the FITC molecules at the nanoslit center such that supersaturation occurred as observed in the fluorescence image at $t = 60$ min, which corresponds to the blurry shape in Fig. 1e. After 1 h of rehydration (RH ~95%), the concentrated FITC molecules diffused into both the source–drain channels ($t = 90$ min), and the steady-state diffusive mass transport resumed ($t = 120$ min). It is noteworthy that this work selected water as the solvent and PDMS as the nanoporous nanochannel material to control the mass transport of aqueous salts. Of course, another combination of solvents including target salts and nanoporous nanochannel materials may exist, requiring additional fundamental research efforts.

**Transport-control characterization with nanoslit lengths.** The transport of small molecules along the nanoslit was quantified for various global humidity conditions (Fig. 2). Phosphate-buffered saline (PBS) solutions with and without FITC were loaded into the source and drain channels, respectively. The FI of the test chamber was measured and the transient transport rate of the FITC molecules through the 800-μm-long nanoslit was quantified (Fig. 2a and Supplementary Fig. 1). First, solvent evaporation through the nanoslit walls was inhibited by using a fully hydrated GECMN and maintaining high humidity (RH ~95%)[27], such that the hydrated GECMN continuously exhibited a constant FI at the test chamber (Supplementary Fig. 1b). Second, the dehydrated GECMN (RH ~20%) prevented the FITC molecules from penetrating the nanoslit, such that FI was not detected in the test chamber (Supplementary Fig. 1c). Third, the rehydrated GECMN was initially dehydrated at low humidity and both channels were filled with PBS solution ($t < 0$ min); subsequently the rehydrated GECMN was exposed to high humidity and the source channel was injected with FITC solution ($t = 0$ min). In this case, the FI

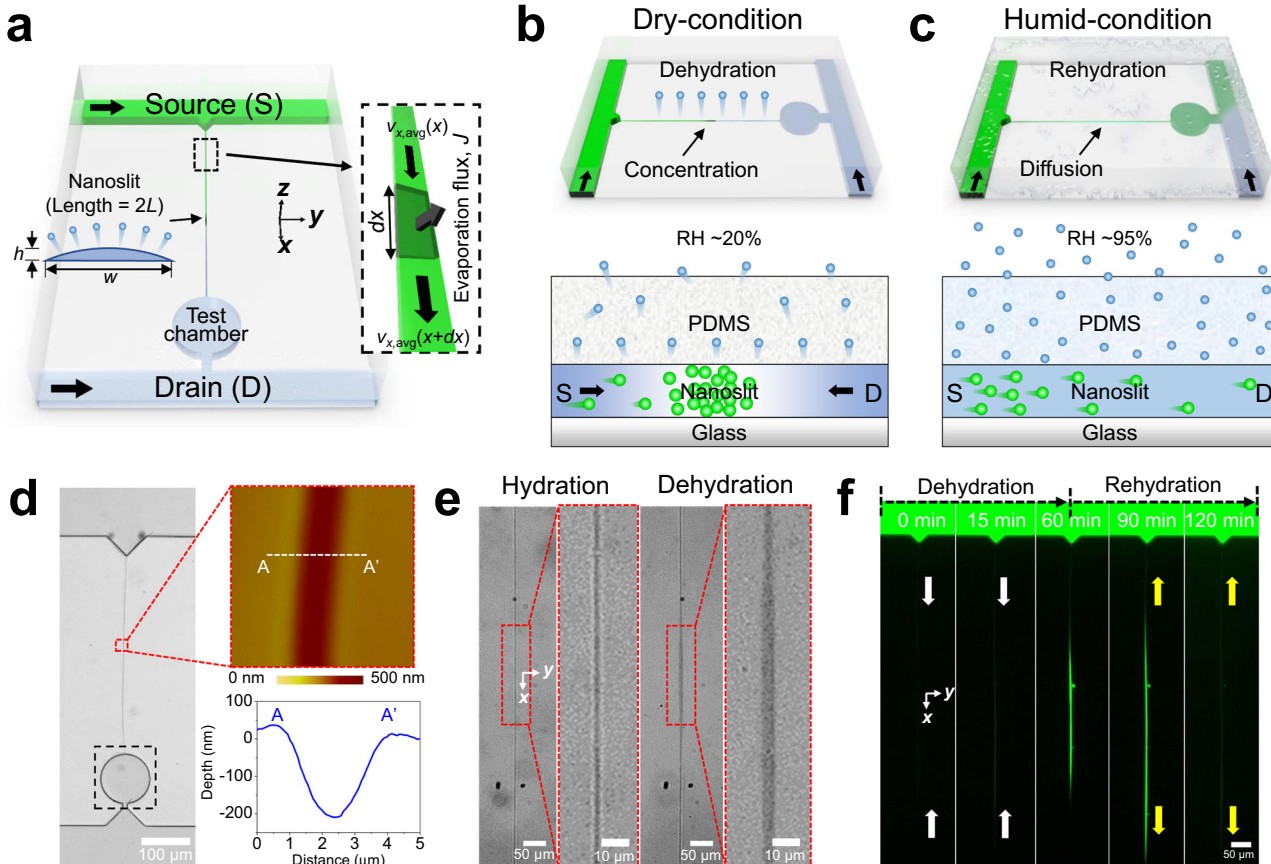

**Fig. 1 Global, evaporation-controlled, micro/nanofluidic device (GECMN) for transport control of small molecules. a** Schematic illustration of the GECMN consisting of two microchannels connected to the nanoslit. **b** Diffusive mass transport of small molecules toward the drain channel is prohibited by the evaporation-driven advective flow from the drain toward the center of the nanoslit, making the small molecules accumulate in the dehydrated nanoslit. **c** Accumulated small molecules transport toward the drain channel due to diffusion for the hydrated nanoslit. **d** Microscopic and atomic force microscopy (AFM) images depicting the nanoslit morphology. **e** Microscopic images of the nanoslit at high and low humidity, respectively. **f** Sequential fluorescence images indicating that the fluorescein isothiocyanate (FITC) molecules accumulated at low humidity (relative humidity; RH ~20%) and are transported via diffusion at high humidity (RH ~95%) at 1-h intervals.

increased gradually over time due to the concentrated solute diffusion during the recovery of the hydrated state (Fig. 2a). After sufficient diffusion occurred and advective flow vanished at the fully hydrated state, the FI gradually decreased and plateaued, similar to the hydrated device in Supplementary Fig. 1b ($t = 150$ min, steady-state diffusive transport between the source and drain).

Figure 2b quantified the diffusive transport rate of the FITC molecules by measuring their FIs at the test chamber when the GECMN was fully hydrated and the global environment was maintained at high humidity (RH ~95%). When both channels were filled, a concentration gradient developed and caused the diffusive transport of the FITC molecules. As a result, the FIs at the test chamber, which indirectly indicated the FITC concentration, gradually increased for ~20 min before reaching a steady-state equilibrium. Overall, the FIs significantly depended on the nanoslit length; as the length decreased, the FIs increased. This is because the concentration gradient increased across the nanoslit, increasing the diffusive transport rate.

In Fig. 2c, the temporal humidity conditions of the GECMN were manipulated in the same manner as those in Fig. 2a. The FIs displayed two distinct trends depending on the nanoslit length. For the shorter nanoslits (50, 100, and 200 μm), the FIs increased when the concentration gradient was generated and plateaued after ~20–30 min, similar to the trend observed in Fig. 2b. For the

longer nanoslits (400 and 800 μm), the FIs initially remained undetectable, and then varied as indicated by the black arrows. The time difference between the initial diffusion of the concentrated solutes was related to the diffusion time ($t_D \sim L^2$). Over a long timescale, both solutions plateaued and behaved similarly to those in Fig. 2b. The FITC molecules in the short nanoslits were not affected by evaporation-driven advection, demonstrating that evaporation-driven transport control only occurred for longer nanoslits (≥400 μm). In other words, the evaporation-driven advective transport and the diffusive mass transport competed with and balanced each other, and were dependent on the nanoslit length. During the rehydration process, the FITC molecules accumulated at the center of the nanoslit for initial recovery, and then began to diffuse toward the drain channel across the test chamber. The repeated and reversible switching between dehydration and rehydration was performed for 800-μm-long nanoslits, demonstrating that GECMNs can be used as concentrators, pumps, and valves for small molecules (Supplementary Fig. 2). It is possible to compare the plotted graphs in Fig. 2b and c in a quantitative manner but the similar comparison between Fig. 2 and Supplementary Fig. 2 should be refrained from as the experimental conditions slightly differed. The experimental results of each nanoslit length shown in Fig. 2b and c were obtained using a single device and each device having a nanoslit length was repeatedly tested ($n = 3$) to

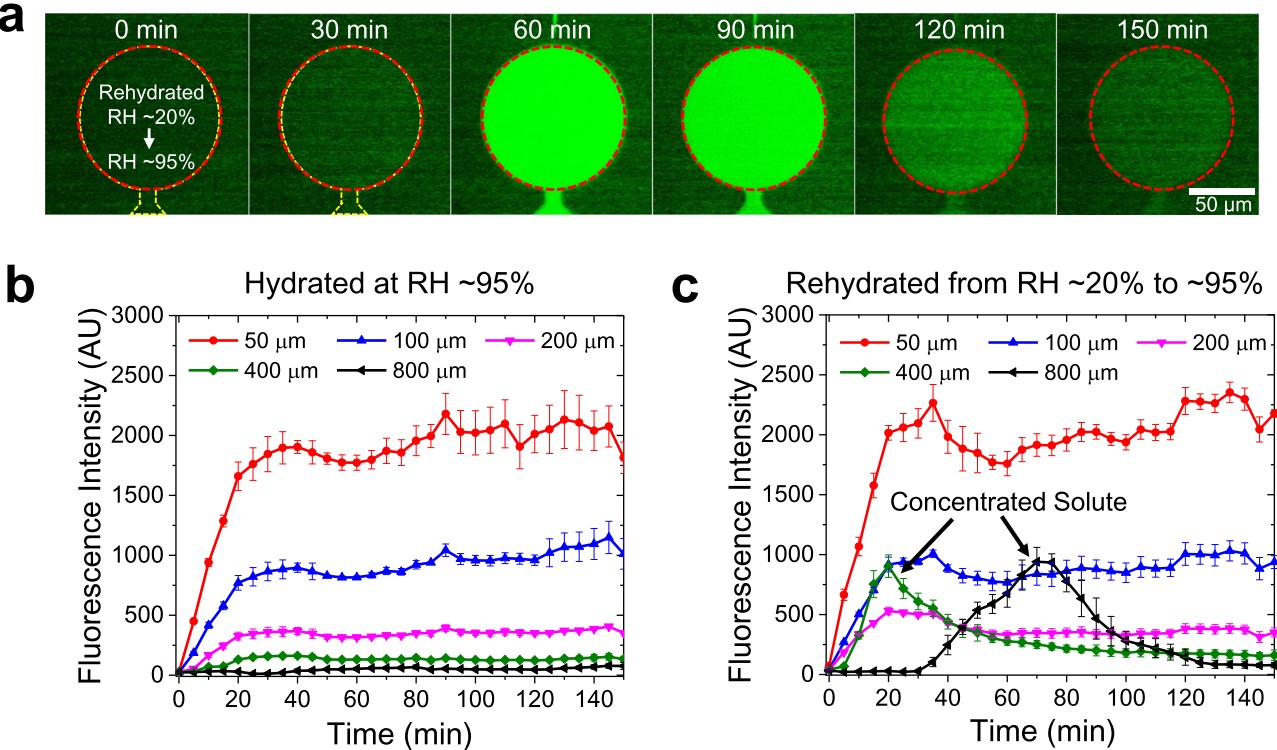

**Fig. 2 Diffusive transport rate of the FITCs along the nanoslit in the GECMN, quantified using varying RH. a** Quantification using the fluorescence intensities (FIs) of the circular test chamber, which are acquired over time. Initially, the GECMN is exposed to an RH of ~20% ($t \leq 0$ min). Then, the RH changes to 95% (after $t = 0$ min) with the injection of the FITC solution, such that the FIs significantly increase. After the evaporation flux is fully prohibited and the concentrated solutes sufficiently diffuse, the FIs decrease. **b** For the completely hydrated condition (RH ~95%), injection of the FITC solution gradually increases the FIs for ~20 min and then plateaus regardless of the nanoslit length. **c** Under the same conditions as those in (**a**), injection of the FITC solution induces the concentration of the solute as shown in the 400 and 800 μm nanoslits but induces a plateau as in (**b**) for the short nanoslits (50, 100, and 200 μm). All error bars represent the standard errors on the means ($n = 3$).

minimize the experimental errors by the nanoslits. To remove the remaining FITC molecules in the used devices, fresh PBS solution was introduced for 1 h after each experiment at high humidity (RH ~95%). For Fig. 2c, the devices were dehydrated at low humidity (RH ~20%) for 1 h after cleaning to secure the same initial condition of the devices.

**Numerical and parametric study on molecular transport**. To further validate the transport-control mechanism, we established a theoretical model based on advection and diffusion and conducted numerical simulations. Figure 3 depicts the numerical simulation results used to determine the solvent evaporation flux and perform parameter studies. First, we applied the continuity equation to the nanoslit to obtain the averaged advective flow velocity[21]:

$$v_{x,\mathrm{avg}}(x) = -\frac{Jx}{\rho h},\tag{1}$$

where $J$ is the solvent evaporation flux through the nanoslit walls, $\rho$ is the solvent density ($\rho = 997$ kg m$^{-3}$ for water), and $h$ is the nanoslit height ($h = 200$ nm). The governing equation for advective and diffusive transport in the 1D nanoslit can be expressed in a nondimensionalized format as follows:

$$\frac{\partial \tilde{c}}{\partial \tilde{\tau}} = \frac{t_o D}{L^2}\frac{\partial^2 \tilde{c}}{\partial \tilde{x}^2} + \frac{t_o J}{\rho h}\tilde{x}\frac{\partial \tilde{c}}{\partial \tilde{x}} + \frac{t_o J}{\rho h}\tilde{c},\tag{2}$$

where $\tilde{c}(\tilde{x}, \tilde{\tau}) = \frac{c(\tilde{x}, \tilde{\tau})}{c_o}$ is the dimensionless solute concentration, $\tilde{\tau} = \frac{t}{t_o}$ is the dimensionless time, $\tilde{x} = \frac{x}{L}$ is the dimensionless distance from the nanoslit center, $c_o$ is the solute concentration in the source

channel, $t_o$ is the dehydration time, and $D$ is the solute diffusivity (see Supplementary Note 1). In Eq. (2), the dynamics of the solute transport were determined by the nondimensionalized diffusion coefficient and evaporation flux, expressed as $\tilde{D} = \frac{t_o D}{L^2}$ and $\tilde{J} = \frac{t_o J}{\rho h}$, respectively. Therefore, the transport control could be described by the properties of the solute/solvent and the dimensions of the nanoslit.

Figure 3a depicts the solute FIs for an 800-μm-long nanoslit after 15 min of dehydration, which is generally required for the nanoslit to fully react to the global humidity. As expected, the FITC molecules accumulated at the center of the nanoslit with a bell-like distribution. Figure 3b illustrates the numerical simulation results obtained from three different dimensional domains (i.e., 1D, 2D, and 3D; see Supplementary Note 2). Owing to the longitudinal structure of the nanoslit (i.e., $L \gg w \gg h$), the 1D model was accurate enough to predict the concentration distribution of the solute ($t_o = 15$ min) when $J = 0.946 \times 10^{-5}$ kg m$^{-2}$ s$^{-1}$ was assumed, and demonstrates good agreement with the experimental results. Figure 3c depicts the transient numerical simulation results when $t_o = 1$ h and $2L = 50, 100, 200, 400,$ and $800$ μm. For the short nanoslits ($2L \leq 200$ μm), diffusion was stronger than advection such that the solute concentration distribution reached the steady state and became approximately linear in a considerably short time. As a result, no solutes concentrated in the nanoslit, instead diffusing along the nanoslit. These results demonstrated good agreement with those depicted in Fig. 2c; the molecules concentrated only for long nanoslits ($2L \geq 400$ μm). For the long nanoslits, as advection increased, the highest concentration of the solutes ($c_{\mathrm{max}}$) was observed near the center ($\tilde{x} = 0$). For the 800-μm-long

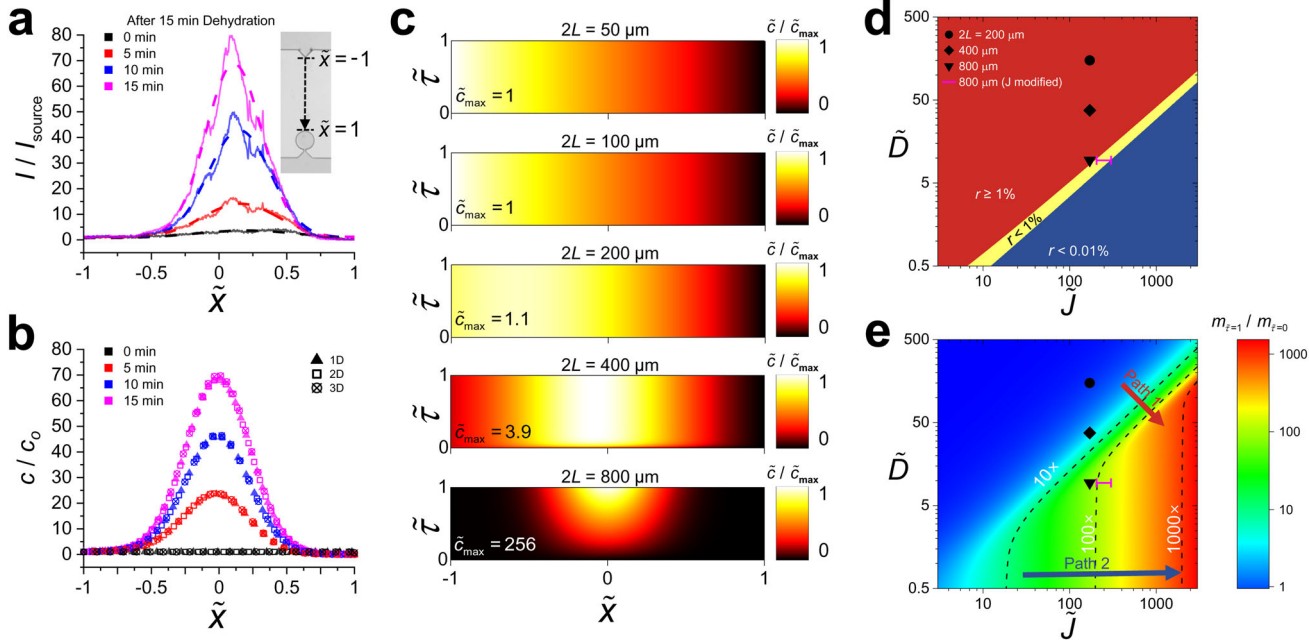

**Fig. 3 Numerical simulation results validating diffusion- and advection-driven mass transport. a** Experimental concentration distribution of the FITCs along the 800-μm-long nanoslit. After 15 min of dehydration ($t = 0$ min), the normalized FIs, $I/I_{source}$ are measured from the nanoslit for 15 min at 5-min intervals. **b** Numerical simulation results of the three different transient models (i.e., 1D, 2D, and 3D) under the same conditions as in the experiment in (**a**). **c** Transient numerical simulation results of the solute concentration distribution for the five different nanoslit lengths (i.e., $2L = 50, 100, 200, 400,$ and 800 μm). The temporal solute concentration distribution along the nanoslit is normalized by the global maximum concentration calculated at $t = t_o$. **d** Two nondimensionalized parameters (i.e., $\tilde{D}$ and $\tilde{J}$) determine the reliability of the transport-control mechanism for small molecules. In other words, the nanoslit actively controls the mass transport of the solutes. **e** Total mass of the solutes concentrated inside the nanoslit (i.e., $m_{\tilde{\tau}=1}/m_{\tilde{\tau}=0}$) is determined and estimated by the two nondimensionalized parameters (i.e., $\tilde{D}$ and $\tilde{J}$). The contour lines separate the parametric conditions to achieve concentration factors of 10-, 100-, and 1000-fold, respectively.

nanoslit, the solutes concentrated in the center, similar to Fig. 1f ($t = 60$ min); refer to Supplementary Movie 1. Although both the long nanoslits concentrated the solute near the center the 400-μm-long nanoslit was not long enough to continuously concentrate the solute, and thus quickly reached the steady state where the solute no longer concentrated inside the nanoslit. Furthermore, we inferred that large and small molecules, which have different diffusion coefficients, can also be controlled by modifying the length and evaporation rate of the nanoslits. In particular, large molecules show slower diffusive transport than small ones. Therefore, shorter nanoslits are more advantageous than longer ones from the viewpoint of the diffusive transport time along the nanoslit, which is directly related to net assay time on a chip.

We further characterized the simulation results using two nondimensionalized parameters. Figure 3d depicts the parametric conditions of the nanoslit for mass transport control of the solutes. For example, the two nondimensionalized parameters of the 800-μm-long nanoslit for FITC were $\tilde{D} = 9.38$ and $\tilde{J} = 170.79$, and their ranges were 0.5–500 and 3–3000, respectively. In addition, we defined a criterion $r$, the ratio of the solute diffusion migrating toward the drain channel to the solute advection originating from the source channel at $t = 1$ h (i.e., $r = -\tilde{D}\frac{\partial \tilde{c}}{\partial \tilde{x}}\big|_{\tilde{x}=1} \times \left(\tilde{J} \times \tilde{c}(\tilde{x} = -1)\right)^{-1}$ at $\tilde{\tau} = 1$). As a result, the parametric conditions for the blue region (Fig. 3d) indicated that <0.01% of solutes that entered the nanoslit by advection $\left(\tilde{J} \times \tilde{c}(\tilde{x} = -1)\right)$ diffused toward the drain channel $\left(-\tilde{D}\frac{\partial \tilde{c}}{\partial \tilde{x}}\big|_{\tilde{x}=1}\right)$, allowing transport control with tight transport gating ($r < 0.01\%$). In contrast, the parametric conditions for the yellow region indicated that the transport gate worked with a small

amount of leakage ($r < 1\%$), and those for the red region did not guarantee transport gating by the nanoslit ($r > 1\%$), demonstrating the low reliability and accuracy of transport control in solutes. For example, with a small $\tilde{D}$ and a large $\tilde{J}$, the nanoslit worked perfectly as a gate for the transport control of solutes and vice versa. The difference in the solute transport behavior can be easily analyzed by using the nondimensionalized Péclet number: $\text{Pe} = \frac{v_{x,\text{avg}}(x=L)L}{D} = \frac{JL^2}{\rho h D}$, where $D = 0.417 \times 10^{-9}$ m$^2$ s$^{-1}$ for FITC[28]. The line at $r = 1\%$ corresponded to a low Pe $\left(\frac{\tilde{J}}{\tilde{D}}\right) = 26.1$ whereas the line at $r = 0.01\%$ corresponded to a high Pe = 35.7. Figure 3e estimates the concentration factor of solutes at the nanoslit. The total mass of the solute was calculated using $m = \int_{-1}^{1} \tilde{c} d\tilde{x}$ with a color scale indicating the concentration factor ($m_{\tilde{\tau}=1}/m_{\tilde{\tau}=0}$), which is the total solute mass in the nanoslit at $\tilde{\tau} = 1$ normalized by the initial total solute mass ($m_{\tilde{\tau}=0} = 2$). The contour lines separated the parametric conditions to achieve concentration factors of 10-, 100-, and 1000-fold, respectively. The concentration factors increased drastically along Path 1 (indicated by the red arrow; Fig. 3e) crossing the yellow gating border in Fig. 3d. They also increased linearly along Path 2 (indicated by the blue arrow; Fig. 3e), inside the blue region in Fig. 3d, implying that gating worked reliably under these conditions. This result will aid the design and fabrication of a small molecule pre-concentrator on a chip for portable biosensors and biochips[29,30].

**Local evaporation-control.** We also demonstrated a local, evaporation-controlled, micro/nanofluidic device (LECMN) for independent and addressable gating through local manipulation

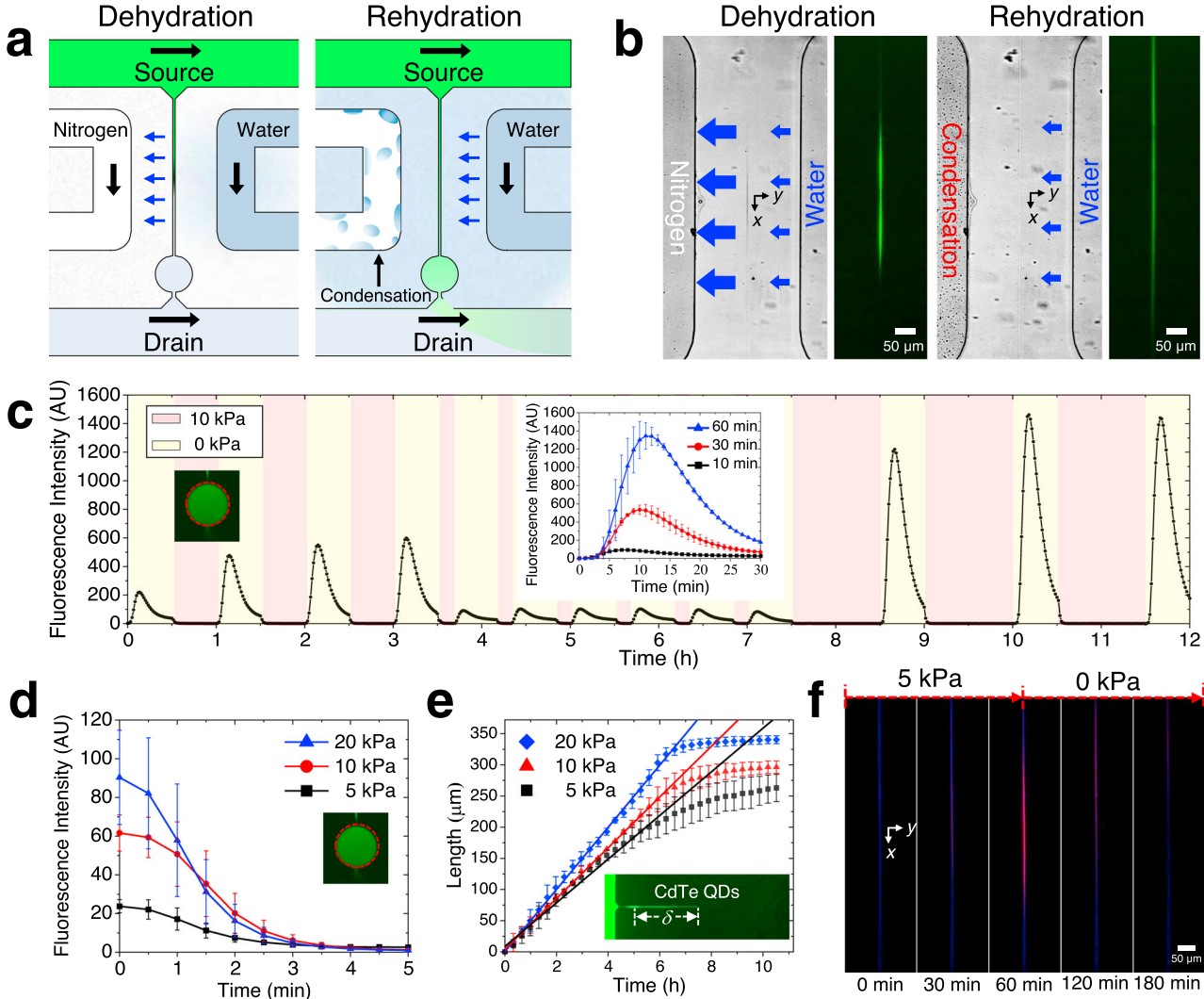

**Fig. 4 Local, evaporation-controlled, micro/nanofluidic device (LECMN), enabling addressable molecule transport gating in a micro/nanofluidic device. a** Schematic of individual molecule gating in the nanoslit by integrating two neighboring microchannels to supply the nitrogen gas and water, respectively. The blue arrows near the nanoslit depict the evaporation of water through the walls. **b** Microscopic images of the micro/nanofluidic device during dehydration (10 kPa) and rehydration (0 kPa), which are actively switched using the dehydrating microchannel. **c** The FIs of the test chamber are continuously measured over time, demonstrating that the mass transport of the solutes along the nanoslit can be controlled by modulating the duty cycles of dehydration and rehydration. The inset depicts the FIs of the FITC molecules transported through the test chamber during the rehydration period after three different dehydration periods at 10 kPa. **d** Quantification of the time required to wash the solutes remaining in the test chamber during dehydration with various pressures applied to the dehydrating microchannel. **e** Quantum dots (QDs) loaded into the source channel are accumulated and then assembled to produce a nanowire-like structure from the center of the nanoslit, as shown in the inset. The growth rate of the QD-structure is affected by the evaporation flux and manipulated by the pressure applied to the dehydrating microchannel. **f** A mixture of 200 μM of acriflavine (fluorescent blue) and 20 μM of sulforhodamine B sodium (fluorescent red) is introduced to the source channel to investigate the effect of molar weight on the transport phenomenon. All error bars represent the standard errors on the means ($n = 3$).

of the evaporation flux (Fig. 4). We designed a micro/nanofluidic channel network in which two supporting microchannels (a dehydrating and rehydrating microchannel) were integrated into GECMN, where the additional microchannels were parallel to the nanoslit with a spacing of 60 μm (Fig. 4a). The dehydrating microchannel was connected to a pressure regulator of nitrogen gas and the rehydrating microchannel was filled with water (see Methods for details). In addition, we demonstrated that the dehydration and rehydration conditions could be easily altered in the LECMN (Fig. 4b and Supplementary Fig. 3). In fact, we have tested many different experimental conditions/configurations (i.e., with/without nitrogen, with/without water, and global humidity conditions) and then came to a conclusion that a dehydration condition can be easily and rapidly achieved

regardless of the water supply into the rehydrating microchannel and the global humidity condition. On the other hand, both the rehydrating microchannel and the global humidity played a decisive role in rapidly rehydrating the device. From these experiments, we optimized the most rapid and most reliable dehydration and rehydration conditions as shown in Fig. 4. This is why we filled the right rehydrating microchannel with water and controlled the nitrogen gas pressure for the left dehydrating microchannel for the entire period of experiments.

The two microscopic images displayed an optical difference at the center of the nanoslits, showing good agreement with the conceptual mechanism in Fig. 4a (refer to Fig. 1e). During dehydration (when the nitrogen gas flowed into the dehydrating microchannel), both the increase in the FIs and the enlargement

of the dark region at the middle of the nanoslit demonstrated the continuous inflow of the FITC molecules (Supplementary Fig. 3). The dark signal was generated because the local FITC concentration significantly increased and aggregated; this was supported by the FI experiment in Supplementary Fig. 4 and Supplementary Movie 2, wherein the FIs were significantly affected by the concentration of FITC at saturation. However, after rehydration (when the nitrogen gas was stopped by the pressure regulator), the FIs completely recovered and diffused toward the nanoslit sides. After 30 min, the concentration distribution of the FITC molecules along the nanoslit became linear, implying a steady-state diffusive transport. Furthermore, to validate the reusability of the device at severe experimental conditions, we introduced a very high concentration (1 mM) of the FITC solution in the source channel for 3 h under the dehydration condition. After dehydration, severe aggregation occurred inside the nanoslit (Supplementary Fig. 5a). However, the aggregated FITC molecules, which can be crystallized although depending on test molecules and experimental conditions, dissolved rapidly when the condition was switched to rehydration. After 1 h of rehydration, the channel became clean with no aggregated/crystallized FITC molecules remaining (Supplementary Fig. 5b); refer to Supplementary Movie 3. We also demonstrated that an array of three nanoslits could be selectively and independently controlled by integrating a pair of supporting microchannels near each nanoslit (Supplementary Fig. 6). As an example of a practical application, the chemical environment of a target chamber connected to three different source channels through the nanoslit array could be varied on demand by blowing nitrogen gas.

As depicted in Fig. 4c and Supplementary Movie 4, we also demonstrated programmable mass transport control by switching between dehydration and rehydration periodically. For dehydration over various time periods, we fixed the applied pressure at 10 kPa to maintain the evaporation flux while rehydration was only allowed for 30 min. The FIs at the test chamber demonstrated that the mass transport rate along the nanoslit was well controlled, allowing manipulation of the timing and number of small molecules delivered to the target chamber. Therefore, the evaporation-based nanoslit worked as a molecule-valve in the micro/nanofluidic device. The inset depicts the temporal FIs of the test chamber for 30 min of rehydration after dehydration for 10, 30, and 60 min, respectively. This result confirmed that the number of small molecules was accumulated in a controllable manner and then delivered to a target chamber through the nanoslit, thereby demonstrating a small molecule pump in the micro/nanofluidic device. The concentration of small molecules was approximately linearly proportional to the dehydration time when considering the initial 5 min response time.

We further investigated the effect of evaporation flux on transport control by regulating the applied pressure to the dehydrating microchannel (Fig. 4d, e). Dehydration and rehydration were repeated thrice and the FIs of the test chamber were measured (Fig. 4d). We repeated the dehydration and the rehydration every 5 min by applying various pressures to the dehydrating microchannel. Evacuation of the FITC molecule in the test chamber took 4–5 min regardless of the concentration of the accumulated molecules. This indicated that the response times of the LECMN were significantly reduced in comparison with those of the GECMN. All experiments depicted in Fig. 4d were performed three times using a single device.

To further characterize the evaporation flux for the same applied pressure conditions as above, we introduced a quantum dot (QD) suspension solution for the source channel. The solution was adjusted to a pH of 12 to prevent the aggregation of QDs[31]. Since QDs have a considerably larger hydrodynamic radius (~2 nm) than

that of FITC molecules (~0.6 nm), advective flow in the nanoslit was dominant for the QDs. After the QD suspension solution was introduced, the nanowire-like assembly of the QD (NAQD) continuously grew from the center of the nanoslit, demonstrating a linear relationship between the assembly length, $\delta$, and $t$ (Fig. 4e and Supplementary Movie 5). Therefore, $v_{x,\mathrm{avg}}(x)$ was also constant over time and, in turn, the total evaporation rate throughout the nanoslit was constant. Although the aggregation of the QDs was prevented at high pH values, the particles overcame the energy barrier (based on the Derjaguin–Landau–Verwey–Overbeek theory) and were assembled by the advection flow[32,33]. All experiments depicted in Fig. 4e were performed three times ($n = 3$) using new devices for each experiment to consider the device-to-device errors. The NAQD sample was transferred from a glass substrate to an optical adhesive material (Noa 63) to demonstrate that the assembled structure was closely packed as shown in the scanning electron microscopy (SEM) images (Supplementary Fig. 7). To clearly demonstrate the packing structure, we generated NAQDs on a piece of Si wafer under the same conditions as the glass substrate and then obtained transmission electron microscopy (TEM) images (Supplementary Figs. 8 and 9), which ascertained again the same close-packed result as the SEM images. Furthermore, owing to the negligible diffusion of the QDs during the growth of the NAQD, the evaporation flux was calculated using Eq. (1) and the following equation was obtained:

$$f \cdot v_{x,\mathrm{avg}}(-L) = \alpha \cdot G, \qquad (3)$$

where $f$ is the volumetric fraction of the QDs in the solution, which was calculated to be $f = 2.849 \times 10^{-4}$, $\alpha$ is the packing ratio of the QDs, which was 0.64, assuming random close packing of a monodisperse hard sphere[34], and $G$ is the growth rate of the NAQD in Fig. 4e. Using Eq. (1), we calculated the modified evaporation flux from $1.155 \times 10^{-5}$ to $1.644 \times 10^{-5}\,\mathrm{kg\,m^{-2}\,s^{-1}}$ by controlling the pressure applied to the dehydrating microchannel. The nondimensionalized parameters, $\tilde{J}$ and $\tilde{D}$, are indicated by the magenta ranged bar symbol in Fig. 3d and e, resulting in $r < 1\%$ for 5 and 10 kPa, and $r < 0.01\%$ for 20 kPa. Thus, for 20 kPa, <0.01% of the solutes diffused toward the drain channel while the rest either diffused toward the source channel or concentrated inside the nanoslit. However, at 5 kPa, equilibrium was achieved between gating and non-gating. This implied that, depending on their molecular mass (MM), which is negatively correlated with the diffusion coefficient, the small molecules exhibited varying behavior at 5 kPa. Therefore, LECMNs can be used as a molecule-filter for molecules with different MMs in a micro/nanofludic device.

To demonstrate the molecule-filter, 200 and 20 μM acriflavine was introduced in the source and drain channels, respectively, for 1 h with 5 kPa applied to the dehydrating microchannel ($t < 0$ min) (Fig. 4f; Supplementary Movie S6). The acriflavine molecules (MM = 259.74 g mol$^{-1}$) fluoresced in the blue region of the spectrum. First, the acriflavine molecules did not concentrate significantly inside the nanoslit because of their higher diffusion coefficient. However, at 20 kPa they concentrated inside the nanoslit (i.e., higher evaporation flux) (Supplementary Movie S7). Second, a mixture of acriflavine (200 μM) and sulforhodamine B sodium (20 μM, MM = 580.65 g mol$^{-1}$) was introduced in the source channel ($t = 0$ min); the latter molecules fluoresced in the red region of the spectrum. After 1 h at 5 kPa, only the sulforhodamine B sodium molecules concentrated in the nanoslit ($r = 0.6\%$) while the acriflavine molecules diffused toward the drain channel ($r = 13.6\%$). The calculated $r$ implies that at 5 kPa, a larger amount of acriflavine was transported toward the drain channel than that of sulforhodamine B sodium ~22 times when their concentrations on the source channel were the same. After 1 h, the applied pressure was reduced to 0 kPa and maintained for 2 h allowing diffusion of

the small molecules toward the drain. These results are well supported by the numerical simulations in Fig. 3d and e.

To summarize, we detailed how solute transport along a nanoslit can be significantly affected by evaporation at nanoscopic interfaces. Evaporation-driven advection controls the transport of small molecules in the nanoslit while competing with diffusion. We experimentally characterized the transport of small molecules along the nanoslit, which was largely governed by the evaporation flux and nanoslit length. We theoretically supported the experimental results with the advection and diffusion model via numerical simulations, enabling description of the transport with the nondimensionalized diffusion coefficient and evaporation flux. These nondimensionalized parameters, which were easily manipulated by dehydration and rehydration, provided theoretical criteria for the reliability of the transport control and the concentration factors of the target solutes. Furthermore, we characterized the evaporation rate by measuring the growth rates of the NAQD in the nanoslit, allowing small molecules to be separated by their MM. Consequently, we demonstrated that evaporation-driven transport control in nanoslit-based micro/nanofluidic devices can be used as a molecule-valve, -concentrator, -pump, and -filter, showing remarkable potential for a variety of applications in micro/nanofluids. In particular, the working principle can facilitate drug delivery and biosensing research on a chip. In addition, the selective transport control of ionic molecules according to their MM may be practical and powerful for energy conversion studies. Last, molecule concentrations can be used for water purification and desalination, enabling additional application.

## Methods

**Fabrication of micro/nanofluidic devices**. A hybrid-scale micro/nanofluidic device mold was produced by crack-photolithography[23,24]. Detailed dimensions of the microstructures are depicted in Supplementary Fig. 10. Negative patterns produced by crack-photolithography were replicated using a polyethylene terephthalate (PET) film and polyurethane acrylate (PUA) solution. Oxygen inhibition during photocuring of the PUA solution was minimized by adopting a custom-made nitrogen chamber. We introduced a PDMS composite with a bi-layer structure of x-PDMS and PDMS to resolve the roof-collapse problem[35]. The prepared x-PDMS solution was used to spin-coat the PUA mold at 2000 rpm for 1 min, followed by heating at 65 °C for 5 min in the oven. Then, regular PDMS was poured onto the x-PDMS layer and cured at 65 °C for at least 4 h. The dimensions of the PDMS device were fixed at 30 mm × 25 mm × 4 mm (length × width × height) for each experiment. Subsequently, oxygen plasma (20 sccm, 50 W for 30 s) was introduced to treat the x-PDMS surface and a glass substrate for producing hydrophilic surfaces with strong bonding.

**Sample preparation and device handling**. The FITC solution was prepared by dissolving FITC salt into a PBS solution, wherein the final concentration was 50 μM, which was used for imaging the concentration distribution in the experiments depicted in Figs. 1f and 4b. For each FI-analysis experiment in the test chamber, 500 μM of the FITC solution was used. The hydrophilic CdTe QD was prepared with filtered deionized (DI) water (pH = 12) to prevent severe aggregation in the microchannels[31]. Acriflavine and sulforhodamine B sodium solutions were prepared by dissolving the salts into filtered DI water. The fabricated device was initially soaked in water for at least 24 h to make it fully hydrated. All solutions were handled with a syringe pump to maintain the flow rate and hydrostatic pressure during the desired working time. The global RH was controlled within the cell incubator integrated in an inverted microscope system. We employed a pressure regulator connected with nitrogen gas and humid gas for dehydration and rehydration, respectively. The RH was constantly measured using a humidity sensor. For the local, evaporation-controlled, micro/nanofluidic device (LECMN), as illustrated in Fig. 4a, the left microchannel (i.e., the dehydrating microchannel) was connected to a nitrogen gas tanker through a pressure regulator while the right microchannel (i.e., the rehydrating microchannel) was connected to a water reservoir to supply liquid water. Both the dehydrating and rehydrating microchannels enabled to the active and local control of the evaporation flux through the nanoslit walls. The dehydration was generated on demand by applying an arbitrary pressure through the pressure regulator. Conversely, the rehydration was also simply produced by turning off the pressure regulator, stopping the supply of nitrogen gas. With the help of the rehydrating microchannel, the response time for the rehydration was as quick as that for the dehydration. During the rehydration, vapors condensed in the dehydrating microchannel, verifying that the nanoslit was completed rehydrated. The global environment was maintained at high humidity

(i.e., RH ~95%) to remain constant between each experiment. Details of the experimental setup are illustrated in Supplementary Fig. 11.

**Materials and reagents**. A negative photoresist made of SU-8 (SU-8 2010, MicroChem, Newton, MA) was used to produce the master mold. PUA (MINS-311RM) and a PET film (Minuta Tech, Osan, Gyeonggi, Korea) were adopted to replicate the negative micro/nanopatterns of the master mold. The replicated positive PUA mold was silanized with chlorotrimethylsilane (Sigma-Aldrich, Yongin, Gyeonggi, Korea). A regular PDMS (Sylgard 184 silicone elastomer kit, Dow Corning, Midland, MI, USA) and an extra-hard PDMS (x-PDMS) were adopted to produce the micro/nanofluidic device. All chemicals used to prepare x-PDMS were purchased from JSI Silicone (Seongnam, Korea), including linear vinyl siloxane (VDT-731), vinyl Q-siloxane (VQX-221), moderator (SIT 7900.0), platinum catalyst (SIP 6831.2LC), and linear hydride siloxane (HMS-501). The FITC molecule (Sigma-Aldrich, Yongin, Gyeonggi, Korea) and additional fluorescence materials were prepared by purchasing commercial products including hydrophilic CdTe QDs with a 520 nm wavelength (plasmaChem, Berlin, Germany), acriflavine, and sulforhodamine B sodium (Sigma-Aldrich, Yongin, Gyeonggi, Korea). Noa 63 (Norland Products Inc, Cranbury, NJ, USA) was prepared for scanning electron microscopy (SEM) of the QD assembly.

**Experimental setup and data analysis**. Optical and fluorescent images were obtained using an inverted fluorescence microscope (IX-71, Olympus, Shinjuku, Japan) equipped with a charge-coupled device camera (Clara, Andor, Belfast, Northern Ireland). Image J (National Institutes of Health, Bethesda, MD, USA) and OriginPro 2020 software (OriginLab Corp., Northampton, MA, USA) were used to quantitatively analyze the FIs of the images. The background FI was subtracted for all quantitative analyses. Atomic force microscopy (AFM; D3100, Veeco, Plainview, NY, USA) was introduced to quantify the dimensions of the nanoscale cracks. The PDMS and glass substrate were bonded through oxygen plasma treatment (Cute-MP, Femto Science, Hwaseong, Korea). A cell incubator (Chamlide TC, Live Cell Instrument, Seoul, Korea) was equipped on the microscope stage to maintain the desired humidity and temperature (25 °C) of the microfluidic device. A solenoid valve array (S10MM-30-24-2, Pneumadyne Inc, Plymouth, MN, USA) was integrated with a custom-made electric circuit and was controlled using LabVIEW (National Instruments Corp., Austin, TX, USA) code. A syringe pump (PHD ULTRA, Harvard Apparatus, Holliston, MA, USA) and a microfluidic pressure control unit (MFCS-EZ, Fluigent, Villejuif, France) were used to control the flow of the samples and pressure of the dehydrating microchannel, respectively. To measure the temperature and RH in the chamber accurately, a temperature-humidity sensor (SHT15, Sensirion, Anyang, Gyeonggi, Korea) was integrated with a custom-made electric circuit and a commercial data communication system (Arduino Uno R3, Adafruit Industries, New York, NY, USA). The SEM images were obtained using field emission SEM (S-4800, Hitachi, Tokyo, Japan). The TEM images were obtained using field emission TEM (Tecnai G2 F20 X-Twin, FEI, Eindhoven, Netherlands). TEM samples were prepared using focused ion beam (FIB, Quanta 3D FEG, FEI, Eindhoven, Netherlands) that milled the cross-section of the NAQD sample.

## Data availability
The experimental data that support the findings of this study are available from the corresponding author on reasonable request.

## Code availability
Computer code is available from the corresponding author on reasonable request.

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

## Acknowledgements

All microfabrication processes were performed at the UNIST Central Research Facility Center. This work was supported by a National Research Foundation of Korea (NRF) grant funded by the Korean government (MSIT) (NRF-2017R1A2A1A17069723, NRF-2020R1A2C3003344 and NRF-2020R1A4A2002728).

## Author contributions

The manuscript was written through contributions of all authors. All authors have given approval to the final version of the manuscript. S.S., D.H. and T.K. conceived this work. D.H. fabricated the device. S.S. implemented the experimental setup. S.S. and D.H. conducted all the experiments, and S.S. performed numerical simulations to theoretically describe the nanofluidic transport. All authors wrote and approved the paper.

## Competing interests

The authors declare no competing interests.

## Additional information

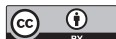

