## [Peer Review File · Nature Communications]

REVIEWER COMMENTS

Reviewer #1 (Remarks to the Author):

This work authored by Seo, Ha and Kim reported fascinating experimental results and offered insightful explanations. Generally, I would like to recommend its publication. However, for the better understanding and more general interest of the readership of Nature Communications, I suggest the authors consider the following suggestions and comments.

1. Apparently, the fabricated devices demonstrate an evaporation-driven transport control strategy. But I am somewhat confused the transport is evaporation-driven or diffusion-driven (concentration gradient).
2. For the future applications, can the authors figure out under what specific circumstances, it is convenient to use relative humidity to control the flow? Difficult for me to image in drug delivery, biosensing, energy conversion, and water desalination.
3. What is the difference, if the authors add a drainage channel from the middle of the nanoslits, which only allows the transport of water? Compared with water evaporation through PDMS, is it more realistic?
4. The inset in Fig. 1a seems weird for me, which looks like another side of the device.

Reviewer #2 (Remarks to the Author):

This is a very unexpected and fascinating manuscript that shows how to control nanofluidic transport of molecules and particles through a process of solvent evaporation. The channels are built out of PDMS that depending on the humidity of the surrounding environment allows solvent evaporation through sides of the channels. The process of evaporation induces the process of advection that competes with diffusion. A detailed description of the mechanism is described and supported by the numerical modeling.

I also very much appreciate the part of the manuscript where a type of ionic and molecular circuit is presented with a possibility to control local transport. The experiments with fluorescent dyes and quantum dots are very convincing.

This is the most exciting and novel manuscript I have read for quite some time and I am happy to recommend it for publication.

Reviewer #3 (Remarks to the Author):

As reviewed the manuscript entitled "Evaporation-driven Transport Control of Small Molecules along Nanoslits", there are couple comments about the motivation, presentation, and experiment provided as below.

1. Motivation.

- a. As the authors stated the study is aiming for broad application, how this concept would work in larger molecules, as most of the cases in life? Could the authors provide preliminary result about that, or maybe comment in the manuscript?
- b. Furthermore, the long dehydration process could induce crystallization in some chemicals and make the nanochannel stuck, how the author could prevent this especially it would affect the transportation in the next step? Do the authors have any evidence that is not the case in the presenting experiments?
- c. I noticed about there are couple relative papers published, but not cited by the authors, such as <https://doi.org/10.1063/1.5137803>, the authors should see if anything you stated that people already have studied on it.

2. Presentation. There are couple comments regarding the unclear parts or missing interpretation.

- a. Interpretation in more detail for figures and videos need to be included, not only the captions. They could be included in the supporting materials if running out of the space in the main article.

The current information provided is not sufficient for readers to understand how the authors perform the experiment and the experiment conditions in multiple experiments.

b. The dimension of nanofluidic channel need to be stated, also if that dimension applied to all experiment devices since it would affect the diffusion phenomena. To have the image inspection is preferrable.

3. Some technical issues and questions about the experiments.

a. In Figure S1b, why the fluorescence intensity is not homogeneous in the test chamber? This figure is confusing compare to other figures, Figure 2a for example, the experiment conditions are not stated as well.

b. In Figure 2b, if the dimension of the test chamber is the same, the molecular capacity should be the same even the nano/microchannel beforehand has the different lengths, time is the only thing matters. So, the dye diffuses in the longer channel length device should still gradually increases till reaching the maximum capacity as 50 μm long case and would not going to the steady state before that. Could the authors comment on this?

c. There is a data confliction between Figure 2c and Figure S2b. As the same 800 μm long device, why it only took around 40 min to reach around 1000 AU in Figure 2c, but it took more than 1 h to reach 300 AU in Figure S2b?

d. In Figure 3b, there are four curves but only three labels on it. In Figure 3c, the color code is confusing due to the C_{max} is different in 5 different length channels, that makes those 5 results not really compatible. The authors claimed this is a good agreement with Figure 2c, however, as an example, the florescence intensity of 50 μm at 1 h is 2 times greater than the florescence intensity of 100 μm in Figure 2c, we don't see this in Figure 3c, if I understand that correctly. It is hard to get the point of the simulation results between 400 μm and 800 μm as well.

e. In Fiugre 4, the authors need to provide the control experiment to clarify the phenomena in Dehydration (with/without nitrogen and water for example) and Rehydration (with/without nitrogen) of Figure 4b.

f. Could the authors comment on how the quantum dot can be assembled themselves, especially the authors had tried to avoid aggregation? From the SEM images in Figure S7, there is no evidence that quantum dot assembled since there are only few of them in the field of view and pretty far away.

g. In Figure S5, could the authors provide a video for this experiment for clarification?

Point by point responses to the reviewers' comments

Referee: #1(Remarks to the Author):

This work authored by Seo, Ha and Kim reported fascinating experimental results and offered insightful explanations. Generally, I would like to recommend its publication. However, for the better understanding and more general interest of the readership of Nature Communications, I suggest the authors consider the following suggestions and comments.

1. Apparently, the fabricated devices demonstrate an evaporation-driven transport control strategy. But I am somewhat confused the transport is evaporation-driven or diffusion-driven (concentration gradient).

Answer: The transport is largely driven by evaporation, but it seems to be confusing as indicated by the reviewer. In fact, we intended that evaporation-driven advection controls the transport of small molecules in the nanoslit while competing with diffusion. To clarify our intention, first of all we modified the title and relevant sentences from "Evaporation-driven transport control" to "Evaporation-driven transport-control". And then, we added a clear sentence; please refer to the lines from 326 to 327 on page 15-16. We believe that the revised title and manuscript help potential readers to avoid any unnecessary misunderstanding.

2. For the future applications, can the authors figure out under what specific circumstances, it is convenient to use relative humidity to control the flow? Difficult for me to image in drug delivery, biosensing, energy conversion, and water desalination.

Answer: Typically, it is of significant importance to control the transport of chemicals and small molecules on a chip because it plays an essential role in a variety of biological and chemical assays. From this point of view, it is advantageous that the nanoslit can be easily replicated from a master mold on a silicon wafer, which can be simply fabricated by the crack-assisted photolithography consisting of a two-step process of general photolithography. In addition, nanoslit-integrated microfluidic devices can be limitlessly reproduced by soft lithography, which is thereby more convenient, reliably, and repeatable compared to other membrane-integrated devices. With these advantages, our device can provide a cheap, fast, and reliable experimental environment for many research fields including drug delivery, biosensing, energy conversion and desalination research. For example, our device facilitates drug delivery and biosensing research on a chip because it can prevent undesired sample damage from external forces caused by electric fields, high pressure, etc., which are indispensable for other devices. Furthermore, selective transport control of ionic molecules according to their molecular weight can be practical and powerful for energy conversion studies. Lastly, molecule concentrations can be used for water purification and desalination, enabling additional application. We added these additional explanations to the conclusion section to clarify our argument (please refer to the lines from 338 to 342 on page 16). Thank you for this useful and review comment.

3. What is the difference, if the authors add a drainage channel from the middle of the nanoslits, which only allows the transport of water? Compared with water evaporation through PDMS, is it more realistic?

Answer: Thanks for the clever suggestion. As far as we understood, the reviewer's suggestion seems to place an additional drainage channel near the center of the nanoslit and then connect them (i.e., drainage and nanoslit channels) where a membrane needs to be integrated in order to only transmit water but filter molecules. We expect that the reviewer's suggestion (i.e. the former) can produce similar experimental results as our current ones (i.e. the latter) given that such a membrane

structure causes only advection from the center of the nanoslit to the drainage channel newly added. However, the former results are not exactly the same as the latter ones because the flow velocity for the former is constant along the entire nanoslit whereas that for the latter varies along the nanoslit and depends on the nanoslit length.

In fact, the usage of such a membrane structure is pretty much the same concept as ours because PDMS only allows the penetration of water, not that of small molecules. The only difference is the way that water molecules get out of the nanoslit wall; that is, pressure for the former (i.e., reviewer's suggestion) whereas evaporation for the latter (i.e., our cases). According to the reviewer's suggestion, we need to push water molecules through the membrane by applying a very strong pressure to overcome the osmotic pressure across the membrane. The pressure is required to create a flow velocity of $\sim 30 \text{ } \mu\text{m/s}$, which is generated along the nanoslit in our current device. Therefore, we believe that our current device concept shows more realistic advantages. In addition, we would like to point out that such pressure generation needs an external energy source or equipment, which results in that the target sample can be damaged by such an energy source in an undesired manner. However, our device completely eliminates the need of additional energy sources and equipment. Moreover, for local transport control, the device suggested by the reviewer may be more complicated because each nanoslit needs an independent design for membranes with its own energy sources. Of course, the former and the latter may have their own pros and cons. In sum, we do appreciate the reviewer's suggestion, which has helped us to improve our device for potential application.

4. The inset in Fig. 1a seems weird for me, which looks like another side of the device.

Answer: We have just noticed that the inset in Fig. 1a looks like either another side of the device or another connection between the source and drain channels. We modified the insets for clarity as follows.

Reviewer #2 (Remarks to the Author):

This is a very unexpected and fascinating manuscript that shows how to control nanofluidic transport of molecules and particles through a process of solvent evaporation. The channels are built out of PDMS that depending on the humidity of the surrounding environment allows solvent evaporation through sides of the channels. The process of evaporation induces the process of advection that competes with diffusion. A detailed description of the mechanism is described and supported by the numerical modeling.

I also very much appreciate the part of the manuscript where a type of ionic and molecular circuit is presented with a possibility to control local transport. The experiments with fluorescent dyes and quantum dots are very convincing.

This is the most exciting and novel manuscript I have read for quite some time and I am happy to recommend it for publication.

Answer: We would like to thank the reviewers for the positive review comments on our manuscript. In fact, we are preparing an application paper using the local transport control of a nanoslit array. We hope the reviewer will be able to read it in the near future. Thank you again.

Reviewer #3 (Remarks to the Author):

As reviewed the manuscript entitled "Evaporation-driven Transport Control of Small Molecules along Nanoslits", there are couple comments about the motivation, presentation, and experiment provided as below.

1. Motivation.

a. As the authors stated the study is aiming for broad application, how this concept would work in larger molecules, as most of the cases in life? Could the authors provide preliminary result about that, or maybe comment in the manuscript?

Answer: Generally, the larger the molecule, the larger the hydrodynamic radius. Therefore, large molecules have smaller diffusion coefficients. As a result, large molecules have an advantage in accumulation in the nanoslit during dehydration. However, large molecules show slower transport than small molecules during rehydration when diffusion becomes more dominant than advection (i.e., evaporation-driven transport). From this reason, we can see that the nanoslit length needs to be as short as possible to reduce the diffusion time for dealing with larger molecules using our device. In conclusion, in order to deal with larger target molecules, the device should be designed by taking into consideration both the nanoslit length and the evaporation flux, which are determined by the diffusion coefficients (refer to Fig. 3d and 3e) and assay purposes of target molecules. However, we agree that the reviewer's inquiry should be addressed in the manuscript for potential readers so that we added some part of this explanation and we added the remaining part regarding the application to the result section. Please check the lines from 180 to 185 on page 9 and those from 338 to 342 on page 16.

b. Furthermore, the long dehydration process could induce crystallization in some chemicals and make the nanochannel stuck, how the author could prevent this especially it would affect the transportation in the next step? Do the authors have any evidence that is not the case in the presenting experiments?

Answer: This should be a very sharp and critical comment. In fact, we have observed the aggregation of FITC molecules during our experiments but we are not sure whether their crystallization occurred or not. As shown in Fig. 4b and Supplementary Fig. 3a (i.e., the dehydration experiments), the optically dark regions indicate that the aggregation/crystallization of FITC molecules has occurred so that they lost their fluorescence signal. However, as the reviewer mentioned, such extreme cases need to be further verified. To clarify this important point, we performed another experiment. That is, we dehydrated the device by applying 20 kPa of pressure and introducing a very high concentration of FITC solution (1 mM) to the source channel for 3 h, which induced severe aggregation/crystallization conditions inside the nanoslit. As a result, the fluorescence property of FITC molecules disappeared. And then, we rehydrated the device and observed that the aggregated/crystallized FITC molecules dissolved again and diffused rapidly as shown in Supplementary Fig. 5 and Supplementary Movie 3 in the revised manuscript (please refer to the lines from 241 to 249 on page 12 and in Supplementary information, refer to the lines from 135 to 140 on page 11).

Repeatedly speaking, we studied only water-soluble molecules. Instead, nonpolar solvents can be employed in the exact same manner as water to convey water-insoluble small molecules while they evaporate from the nanoslit through the PDMS walls to an ambient environment. However, it is noted that other solvents are not compatible with PDMS because they are significantly absorbed by PDMS; typically, the PDMS walls need to be treated with CYTOP™ or Teflon to prevent absorption and geometrical deformation although such surface treatment significantly deteriorates evaporation. Therefore, we chose water as solvent and water-soluble molecules as solute to focus on the transport of small molecules by reducing the complexity derived from the relations between solvents and nanoporous nanochannel materials. In conclusion, such an aggregation or crystallization may

depend on intrinsic chemical properties, unfortunately appearing to be out of the scope of this work and requiring an additional study in the future. We added this explanation to the introduction (please refer to the lines from 92 to 95 on page 5).

c. I noticed about there are couple relative papers published, but not cited by the authors, such as <https://doi.org/10.1063/1.5137803>, the authors should see if anything you stated that people already have studied on it.

Answer: We have read the references recommended by the reviewers and found that they reported fundamentals of evaporation through nanopores. We cited the paper and another paper relevant to our work to the introduction as suggested (please refer to the line 44 to 45 on page 3). It is fortunate and true that the key mechanism of our device is well supported by classical fundamentals found in fluid mechanics and heat and mass transfer. However, we demonstrated that evaporation-driven advection can be as competent as diffusion-driven transport for small molecules at nanoscales for the first time to the best of our knowledge. In other words, the fundamentals can be the same but the scale and/or dimensional difference can be very novel, showing remarkable potential for micro-/nanofluidic transport control of small molecules on a chip.

2. Presentation. There are couple comments regarding the unclear parts or missing interpretation.

a. Interpretation in more detail for figures and videos need to be included, not only the captions. They could be included in the supporting materials if running out of the space in the main article. The current information provided is not sufficient for readers to understand how the authors perform the experiment and the experiment conditions in multiple experiments.

Answer: Sorry for the insufficient interpretation of the experiments and experimental conditions. We complemented the experimental setup and conditions in the revised Supplementary Information (Supplementary Fig. 10, 11) for better understanding. And, we added more detailed experimental conditions to the revised manuscript (Fig. 2b, 2c, 4d, 4e and Supplementary Fig. 2) as suggested by the reviewer. In addition, as the journal guidelines say, we added the number of experiment we performed and that of devices we used during the each experiment. Lastly, used materials, experimental processes, and customized experimental setup were well described in the method section in the revised manuscript. In more details, please refer to the lines from 137 to 142 on page 7 (Fig. 2b, 2c), those from 269 to 274 on page 13 (Fig. 4d), and those from 285 to 287 on page 14 (Fig. 4e). Besides, in the Supplementary Information, refer to the lines from 115 to 117 on page 8 (Supplementary Fig. 2) and those from 195 to 202 on page 17 (experimental setup).

b. The dimension of nanofluidic channel need to be stated, also if that dimension applied to all experiment devices since it would affect the diffusion phenomena. To have the image inspection is preferable.

Answer: As the reviewer gave a comment, we added the dimensions of our devices to the section titled "Fabrication of micro/nanofluidic devices". We also added the detailed dimensions of the device to the Supplementary Information in Supplementary Fig. 10. Please refer to the line 327 and those from 354 to 355 on page 15-16 in the revised manuscript (dimensions of PDMS device) and those from 187 to 192 on page 16 in the revised Supplementary Information (dimensions of microstructures of the device).

3. Some technical issues and questions about the experiments.

a. In Figure S1b, why the fluorescence intensity is not homogeneous in the test chamber? This figure is confusing compare to other figures, Figure 2a for example, the experiment conditions are not stated as well.

Answer: It appears that the inhomogeneity results from the solution impurities (i.e., artifacts) during the repeated experiments with one device. Such artifacts can either absorb fluorescent molecules or fluoresce by themselves as nonspecifically bound on the glass surface. When a solution was carefully filtered with a syringe filter, we could not observe such artifacts as shown in Fig. R1 that was newly obtained by conducting the similar experiment as Fig. 2a for clarity. Another reason for the inhomogeneity may result from the repeated use of the same device to minimize the device-to-device errors. In other words, one nanoslit was repeatedly used for accurate characterization. In fact, we performed all the experiments for Fig. 2a, Supplementary Fig. 1b, and 1c using one device to directly compare their fluorescence intensities at a fully hydrated condition (e.g., the fluorescence intensities of Fig. 2a after 150 min vs. those of Supplementary Fig. 1b after 30 min at steady state). Because the repeated experiments with one device required several solution exchanges, such the impurities/artifacts might be unnecessarily caused. However, as shown in Fig. 2b and 2c, we quantified the fluorescence intensities over time by excluding the effect of the impurities/artifacts during image processing and confirmed that they matched well at the fully hydrated and steady state as aforementioned (refer to the words underlined).

We found that the experiment condition for Fig. 2a has been already described in the original manuscript but we explicitly matched it with the right figure (i.e., Fig. 2a) in the revised manuscript (please refer to the line 110 on page 6). Thank you for your careful review comment.

Figure R1. **Characterization of the fluorescence intensity of the test chamber.** Time-lapse fluorescence images of a fully hydrated PDMS system are shown when the system was exposed to a high humidity (RH ~95%) to maintain hydration. The FITC molecules were continuously transported along the nanoslit and then arrived at the test chamber.

b. In Figure 2b, if the dimension of the test chamber is the same, the molecular capacity should be the same even the nano/microchannel beforehand has the different lengths, time is the only thing matters. So, the dye diffuses in the longer channel length device should still gradually increases till reaching the maximum capacity as 50 μm long case and would not going to the steady state before that. Could the authors comment on this?

Answer: The fluorescence intensities of the test chamber indirectly represents the mass transport rate along the nanoslit (i.e., diffusion). Because the concentration gradient ($\partial c/\partial x$) is linear and depend on the nanoslit length at hydrated state, there is a relationship between the nanoslit length and the fluorescence intensities of the test chamber. The shorter the nanoslits are (i.e., the higher concentration gradients), the higher the fluorescence intensities are observed in the test chamber (i.e., the higher diffusive transport rate), and vice versa. These experimental results show good agreement with the numerical simulation result for the short nanoslits in Fig. 3c. Please refer to our answer given to the review comment in 3.d below for further clarity.

c. There is a data confliction between Figure 2c and Figure S2b. As the same 800 μm long device, why it only took around 40 min to reach around 1000 AU in Figure 2c, but it took more than 1 h to reach 300 AU in Figure S2b?

Answer: In fact, the experimental conditions of Fig. 2c are not the same as those of Supplementary Fig. 2b. Sorry again for misleading and insufficient explanation about the experimental conditions. We did our best to explain them in more details in the revised manuscript.

First of all, the rehydration in Supplementary Fig. 2b was performed after the dehydration for 2 h (i.e., after the pre-concentration of molecules for 2 h). On the other hand, for Fig. 2c, PBS buffer containing no FITC was loaded into the source and drain channels under a dehydration condition before rehydration (i.e., no pre-concentration). And then, for the moment of transition from the dehydration to the rehydration, FITC solution was injected into the source channel, which is described in the caption of Fig. 2c (please refer to the lines from 106 to 110 on page 5-6 and those from 121 to 122 on page 6). Consequently, for Supplementary Fig. 2b, the dehydration lasted for 2 h before the rehydration began (i.e., $t = 2$ h retardation) whereas, for Fig. 2c, the rehydration started right after the FITC solution was loaded (i.e., $t = 0$ min). From this point of view, the fluorescence intensities in Supplementary Fig. 2b started to increase earlier than those in Fig. 2c. In other words, the pre-concentrated FITC molecule started to diffuse towards the test channel from the center of the nanoslit, so that the concentration gradient of Supplementary Fig. 2b became even steeper than that of Fig. 2c. Differently, the fluorescence intensities of Fig. 2c began to increase only 30 minutes after the condition switched (i.e., from dehydration to rehydration condition) because the molecules started to diffuse from the end of the nanoslit to the test chamber and there were no pre-concentrated molecules at the nanoslit. Although the experimental conditions are different, it is possible to estimate the diffusion time of FITC molecules along the nanoslit by taking them into consideration. That is, the time for the FITC to reach its maximum fluorescence intensities was 70 min for Fig. 2c and 80 min for Supplementary Fig. 2b, respectively, showing a reasonable consistency.

d1. In Figure 3b, there are four curves but only three labels on it. In Figure 3c, the color code is confusing due to the C_{max} is different in 5 different length channels, that makes those 5 results not really compatible.

Answer: Thank you for your careful review comment on Fig. 3b and 3c. First, we modified the figure labels in the revised manuscript for clarify. Second, we displayed time-lapse simulation results in Fig. 3c by normalizing each time-dependent result with the maximum concentration. We added more legends to each plot. We hope they can give sufficient information and prevent unnecessary confusion any longer.

d2. The authors claimed this is a good agreement with Figure 2c, however, as an example, the fluorescence intensity of 50 μm at 1 h is 2 times greater than the fluorescence intensity of 100 μm in Figure 2c, we don't see this in Figure 3c, if I understand that correctly. It is hard to get the point of the simulation results between 400 μm and 800 μm as well.

Answer: Not only the small amount of the evaporation-driven flow rate but also the large dimensional difference between the nanoslit at nanoscales and the test chamber at microscales makes the simulation of the entire device challenging, in terms of time, cost, and accuracy. In addition, for the parametric study, 1D approximation was conducted by focusing on the nanoslit of interest. In this context, we assumed that the concentration of the solution at the drain-channel-sided end of the nanoslit ($x = L$) is zero. This assumption represents a conditional set because the molecules concentrated at the nanoslit originate only from the source channel and not from the drain channel. For our micro/nanofluidic system, the test chamber has a significantly larger cross-sectional area orthogonal to the diffusion direction than that of the nanoslit. Because of this substantial difference in the cross-sectional areas, the concentration gradient drops mostly within the nanoslit to satisfy the continuity of the solute transport. Therefore, even if the concentration in the test chamber is not exactly zero, its value is significantly low in comparison with that in the nanoslit. By separating the nanoslit and the test chamber under this assumption, the concentration of the test chamber represents the mass transport rate from the nanoslit to the test chamber rather than the concentration at the end of the nanoslit ($x = L$). From this point of view, the nanoslits of 50, 100, and 200 μm , wherein the solutes are not concentrated, as depicted in Fig. 3c, show good agreement with those depicted in Fig. 2c. In other words, the same linear concentration gradient was observed for the normalized distance and other concentration cases. The shorter nanoslit has a steeper concentration gradient, and the diffusion into the test chamber increases according to Fick's law. Therefore, as depicted in Fig. 2b, as the length of the nanoslit decreases, the fluorescence intensity

at the steady state increases, and vice versa.

In summary, the fluorescence intensities of the test chamber are a result of the diffusion from the nanoslit to the test chamber; thus, the shorter the nanoslit, the greater the fluorescence intensities at the test chamber. Therefore, we concluded that the results of Figs. 3c and 2b show good agreement. This is why that the test chamber is integrated with the nanoslit to quantify the mass transport rate along the nanoslit.

In particular, we performed numerical simulations on the diffusion along the nanoslit and the test chamber using COMSOL Multiphysics (ver 5.5). The dimensions of the numerical domain are identical with those of the micro-/nanofluidic system described in the manuscript. The normalized concentration, c/c_0 at the end of the nanoslit (i.e., the junction between the nanoslit and source channel) was set to 1, and the c/c_0 at the end of the test chamber was set to 0 (i.e., the junction between the inlet of the test chamber and the drain channel). In the steady-state study, as depicted in Fig. R2a and R2b, the concentration level drops mainly in the nanoslit and the concentration of the test chamber is considerably close to zero, and seems negligible enough, as depicted in Fig. R2c. Furthermore, the volume averaged concentrations of the test chamber were 0.00273, 0.00136, 0.00068, 0.00034 and 0.00017 for 50, 100, 200, 400 and 800 μm , respectively. The results demonstrate a reciprocal relation between the concentration in the test chamber and the nanoslit length, which shows good agreement with the results depicted in Fig. 2b.

We added this explanation to Supplementary Note 1 for better explanation on the relationship between the fluorescence intensities of the test chamber and the diffusion rate along the nanoslit toward the test chamber. And we additionally revised the manuscript to elucidate the consistency between Fig. 3c and Fig. 2c (please refer to the lines from 174 on page 8 and in the Supplementary Information, refer to the lines from 39 to 77 on page 3-5).

Figure R2. **3D numerical simulation results obtained by COMSOL Multiphysics to validate the boundary condition in 1D simulation when only diffusion is considered.** The dimensions of the numerical domain are identical to those of the micro/nanofluidic device structure described in the manuscript. **a** Concentration distribution of the entire numerical domain when only diffusion is considered. **b** Concentration distribution along the nanoslit from the source channel ($x = 0 \mu\text{m}$ in Fig. R2a) to the inlet of the test chamber ($x = 310 \mu\text{m}$ in Fig. R2a); the graph shows the calculation result at $y = 0 \mu\text{m}$ and $z = 0.1 \mu\text{m}$ (i.e., 100 nm above the substrate) as depicted in Fig. R2a. The concentration drops significantly over the nanoslit. **c** The concentration distribution of the test chamber is rescaled from that in **b** to focus only on the test chamber. The level of the concentration is considerably low in comparison with that in the nanoslit.

e. In Figure 4, the authors need to provide the control experiment to clarify the phenomena in Dehydration (with/without nitrogen and water for example) and Rehydration (with/without nitrogen) of Figure 4b.

Answer: This is a good point. In fact, we have tested many different experimental conditions/configurations (i.e., with/without nitrogen, with/without water, and global humidity conditions) and then came to a conclusion that a dehydration condition can be easily and rapidly achieved regardless of the rehydrating microchannel (i.e., with/without water supply) and global humidity (i.e., high or low RH) conditions. On the other hand, both the rehydrating microchannel and the global humidity played a decisive role in rapidly rehydrating the device. From these experiments, we optimized the most rapid and most reliable dehydration and rehydration conditions as shown in Fig. 4. This is why we filled the right rehydration microchannel with water and controlled the nitrogen gas pressure for the left dehydration microchannel for the entire period of experiments. We added this discussion to the revised manuscript (please refer to the lines from 222 to 230 on page 11). Lastly, it is unfortunate that we cannot provide control experimental results in a quantitative manner. However, we are sure that the current experimental condition aforementioned and used in this work is well optimized experimentally.

f. Could the authors comment on how the quantum dot can be assembled themselves, especially the authors had tried to avoid aggregation? From the SEM images in Figure S7, there is no evidence that quantum dot assembled since there are only few of them in the field of view and pretty far away.

Answer: According to the DLVO theory, which is related to the stability of particles in the solution, smaller particles are more prone to aggregate because they have a smaller energy barrier. Thus, to prevent the aggregation of quantum dots (size is ~ 2 nm), we used a quantum dot solution at pH = 12, which increases the electric double layer (EDL) repulsion of quantum dots, resulting in better suspension stability before they assemble inside the nanoslit. However, the DLVO theory also says that particles can overcome the barrier and become even closer than the spacing distance in suspension when the energy barrier is overwhelmed by external forces (advective flow in our case). As a result, two particles can spontaneously contact with each other. This is because the van der Waals attraction force becomes greater than the EDL repulsion force inside the energy barrier. From the theoretical rationale, we concluded that quantum dots were assembled inside the nanoslit because the viscous drag force by advective flow forced the quantum dots to aggregate at the center of the nanoslit. In addition, we found that the assembly is so strong and durable so that it keeps its shape on the glass substrate after several hours of sonification while soaked in acetone.

Please refer the fabrication process found in *Nano Lett.* 2016, 16, 4, 2189–2197 (Fabrication processes are in Supporting information) and Lab Chip, 2016,16, 1072-1080 (one of our previous work). Particles that are initially suspended in buffer solution are self-assembled by evaporation-driven flow at the liquid-air interfaces. The self-assembly of the particles reported by these references show high stability even after the interfaces are removed and then fresh liquid solution was newly loaded. Lastly, we agree with the reviewer that the previous SEM image is not good enough to directly prove the assembly of quantum dots so that we imaged the assembled quantum dots again with TEM after repeating the same self-assembly process on a Si wafer as shown below and Supplementary Fig. 9. Please refer to the lines 283-293 on page 14. Besides, in the Supplementary Information, refer to the lines from 169 to 176 on page 14 and those from 178-184 on page 15. Thank you for your careful review comment.

Figure R3. TEM image of the cross-section of the NAQD. It appears that QDs are well assembled in the nanoslit by evaporation-driven advective flow. The lattice spacings of the CdTe crystal structures in the NAQD are distinguishable from those of Pt. The dashed-circles in red indicate the shape of single spherical QDs while the arrows in red do an assembled structure of QDs in part. The NAQD consists of about 50 QD layer into the depth direction of the image that is orthogonal to the cross-section of the NAQD sample as shown in Supplementary Fig. 8b and 8c.

g. In Figure S5, could the authors provide a video for this experiment for clarification?

Answer: Unfortunately, we did not take a video of the experiment. Because the pressure regulator we used has only two outputs, only a manual way was possible for us to switch the dehydration microchannel into which nitrogen gas flowed. This is why we could not take a real-time video. However, we plan to publish another application paper following this manuscript in which an array of nanoslits will be individually and separately controlled to supply small molecules to the target chamber array (e.g., test chamber array) by developing the same approach shown in Supplementary Fig. 6 in the revised manuscript. Note that the figure numbering has changed after revision (i.e., Supplementary Fig. 5 in the original manuscript became Supplementary Fig. 6 in the revised manuscript). We do appreciate all your comments and suggests that have really helped us to improve the overall quality of our work.

– END –

Reviewers' Comments:

Reviewer #1:

Remarks to the Author:

The response and revision are satisfied and Now I can recommend its publication in the current form.

Reviewer #3:

Remarks to the Author:

As reviewed the authors' responses, the revised manuscript is enormously improved. However, there is a minor issue still as in the authors' response, it will be great if the authors can clarify this. It is a nice work for this study, and hopefully the reviewers' comments can decrease readers' concern in any viewpoint. This study could enrich the field of nanofluidics I believe.

The follow up question as below. As the authors below responses, there are couple confusing interpretation for those two figures even the experiment condition is different. It will be great if the authors could clarify the confliction.

- First, the concentration gradient in Fig. 2c is actually steeper than Supplementary Fig. 2b (Fig. S2b) as time estimation for the maximum intensity as below. Moreover, if we only count to the same maximum intensity as in Fig. S2b, around 325 AU, it only takes 10 – 15 min in the Fig. 2c to reach that intensity. I aware of the authors may count the time period from their experiment, but as a reader, we only can read what the authors presented in the figure.
- If my understanding is correctly, the authors stated there is no preconcentration in Fig. 2c, but the label in the plot looks not like that. Similarly, as the maximum fluorescence intensity in Fig. 2c is around 3 times greater than the maximum fluorescence intensity in Fig. S2b, it seems the data shows the preconcentration happened in Fig. 2c, not Fig. S2b.

c. There is a data confliction between Figure 2c and Figure S2b. As the same 800 um long device, why it only took around 40 min to reach around 1000 AU in Figure 2c, but it took more than 1 h to reach 300 AU in Figure S2b?

Answer: In fact, the experimental conditions of Fig. 2c are not the same as those of Supplementary Fig. 2b. Sorry again for misleading and insufficient explanation about the experimental conditions. We did our best to explain them in more details in the revised manuscript.

First of all, the rehydration in Supplementary Fig. 2b was performed after the dehydration for 2 h (i.e., after the pre-concentration of molecules for 2 h). On the other hand, for Fig. 2c, PBS buffer containing no FITC was loaded into the source and drain channels under a dehydration condition before rehydration (i.e., no pre-concentration). And then, for the moment of transition from the dehydration to the rehydration, FITC solution was injected into the source channel, which is described in the caption of Fig. 2c (please refer to the lines from 106 to 110 on page 5-6 and those from 121 to 122 on page 6). Consequently, for Supplementary Fig. 2b, the dehydration lasted for 2 h before the rehydration began (i.e., $t = 2$ h retardation) whereas, for Fig. 2c, the rehydration started right after the FITC solution was loaded (i.e., $t = 0$ min). From this point of view, the fluorescence intensities in Supplementary Fig. 2b started to increase earlier than those in Fig. 2c. In other words, the preconcentrated FITC molecule started to diffuse towards the test channel from the center of the nanoslit, so that the concentration gradient of Supplementary Fig. 2b became even steeper than that of Fig. 2c. Differently, the fluorescence intensities of Fig. 2c began to increase only 30 minutes after the condition switched (i.e., from dehydration to rehydration condition) because the molecules started to diffuse from the end of the nanoslit to the test chamber and there were no preconcentrated molecules at the nanoslit. Although the experimental conditions are different, it is possible to estimate the diffusion time of FITC molecules along the nanoslit by taking them into consideration. That is, the time for the FITC to reach its maximum fluorescence intensities was 70 min for Fig. 2c and 80 min for Supplementary Fig. 2b, respectively, showing a reasonable consistency.

Main text Fig. 2

Supplementary Fig. 2

Point by point responses to the reviewers' comments

Referee: #1

The response and revision are satisfied and Now I can recommend its publication in the current form.

Answer: We feel very thankful for your positive comment.

Referee: #3

As reviewed the authors' responses, the revised manuscript is enormously improved. However, there is a minor issue still as in the authors' response, it will be great if the authors can clarify this. It is a nice work for this study, and hopefully the reviewers' comments can decrease readers' concern in any viewpoint. This study could enrich the field of nanofluidics I believe.

Answer: First of all, we do appreciate the issue raised and comments for our work. It is absolutely true that the previous review comments given by Referee #3 had helped to improve this manuscript. We made additional efforts to further enrich it in this revision. Thank you very much again.

The follow up question as below. As the authors below responses, there are couple confusing interpretation for those two figures even the experiment condition is different. It will be great if the authors could clarify the confliction.

- a. First, the concentration gradient in Fig. 2c is actually steeper than Supplementary Fig. 2b (Fig. S2b) as time estimation for the maximum intensity as below. Moreover, if we only count to the same maximum intensity as in Fig. S2b, around 325 AU, it only takes 10 – 15 min in the Fig. 2c to reach that intensity. I aware of the authors may count the time period from their experiment, but as a reader, we only can read what the authors presented in the figure.

Answer: The reviewer is right. We made a mistake in calculating the concentration gradient. However, as the reviewer knows, fluorescence intensities are measured in an arbitrary unit (AU), indicating the relative fluorescence intensity. Therefore, AU values cannot be directly compared because they are relative and affected by various experimental conditions such as microscopy setting, magnification ratio, exposure time, intensity of fluorescent lamp, etc. As a result, it is possible to compare the plotted graphs in Fig. 2b and 2c in a quantitative manner but the similar comparison between Fig. 2 and Supplementary Fig. 2 should be refrained from as the experimental conditions for fluorescence imaging slightly differed. In this context, we concede that our argument, "the concentration gradient in Supplementary Figure 2b is steeper than that in Figure 2c" in the previous response letter was wrong but we believe that our description and argument in the manuscript are still appropriate without additional revision. Nevertheless, as advised by the reviewer, we simply added the aforementioned statement for clarity; please refer to the lines from 142 to 144 on page 7. We apologize for the confusion in the previous response letter and appreciate your careful review comment again.

- b. If my understanding is correctly, the authors stated there is no preconcentration in Fig. 2c, but the label in the plot looks not like that. Similarly, as the maximum fluorescence intensity in Fig. 2c is around 3 times greater than the maximum fluorescence intensity in Fig. S2b, it seems the data shows the preconcentration happened in Fig. 2c, not Fig. S2b.

Answer: We noticed that the label in Fig. 2c can cause any misunderstanding that molecules were preconcentrated. For clarity, we fixed the label from 'Preconcentrated solute' to 'Concentrated solute'. Thank you again for your careful review.